# Intelligent Coatings with Controlled Wettability for Oil–Water Separation

**DOI:** 10.3390/nano12183120

**Published:** 2022-09-08

**Authors:** Shumin Fan, Yunxiang Li, Rujun Wang, Wenwen Ma, Yipei Shi, Wenxiu Fan, Kelei Zhuo, Guangri Xu

**Affiliations:** 1School of Chemistry and Chemical Engineering, Henan Institute of Science and Technology, Xinxiang 453003, China; 2School of Chemistry and Chemical Engineering, Henan Normal University, Xinxiang 453007, China

**Keywords:** wettability transformation, superhydrophilic/underwater superoleophobic, superhydrophobic/superoleophilic, separation

## Abstract

Intelligent surfaces with controlled wettability have caught much attention in industrial oily wastewater treatment. In this study, a hygro-responsive superhydrophilic/underwater superoleophobic coating was fabricated by the liquid-phase deposition of SiO_2_ grafted with perfluorooctanoic acid. The wettability of the surface could realize the transformation from superhydrophilicity/underwater superoleophobicity (SHI/USOB) to superhydrophobicity/superoleophilicity (SHB/SOI), both of which exhibited excellent separation performance towards different types of oil–water mixtures with the separation efficiency higher than 99%. Furthermore, the long-chain perfluoroakyl substances on the surface could be decomposed by mixing SiO_2_ with TiO_2_ nanoparticles under UV irradiation, which could reduce the pollution to human beings and environment. It is anticipated that the prepared coating with controlled wettability could provide a feasible solution for oil–water separation.

## 1. Introduction

The rapid industrial development has inevitably resulted in increasing oily sewage, which has threatened the ecological environment and human health [1,2,3]. Oily wastewater contains immiscible oil/water mixtures and miscible emulsion that are difficult to separate [4]. Finding effective methods for water treatment has become a significant challenge and an urgent issue for researchers [5]. Superwetting materials have attracted more attention compared with traditional separation methods for high-efficiency oil/water separation, and low costs [6,7]. The superwetting materials could be categorized into three types: oil-removing [8], water-removing [9,10], and controllable oil/water separation materials [11]. The oil-removing type materials could separate oils from water for their superhydrophobicity/superoleophilicity (SHB/SOI) [12,13,14]. The water-removing type materials could allow water to pass through and prevent oils due to their superhydrophilicity/underwater superoleophobicity (SHI/USOB) [15,16,17]. The controllable surface wettability materials offer great promise for on-demand oil/water separation for their low energy consumption and simple design, proving high selectivity and separation efficiency [18,19].

Various controllable separation materials that could switch their wettability have been fabricated by the manipulation of chemical constituents and surface structure. Dang et al. [20] constructed a pH-responsive surface by dip-coating the copolymer and silica on different materials for oily sewage separation. The coated surface exhibited switchable superhydrophilicity and superhydrophobicity. Zhang et al. [21] fabricated a smart surface with controllable wettability by linking a pH-responsive block copolymer. Li et al. [22] synthesized an intelligent material with controlled wettability by coating polyaniline on different substrates. Zhang et al. [23] fabricated the SHI/USOB fabric based on polypyrrole/silver nanoparticles. The SHI/USOB fabric could be transformed into the SHO/SOI fabric by hydrophobic modification. Yang et al. [24] created a superhydrophobic surface by soft lithography and bio-inspired modification of SiO_2_ nanoparticles. The surface could transform from a superhydrophobic state to a high water-adhesive state after irradiation with UV/visible light. Idriss et al. [25] designed wettability switchable material based on piezo-responsible polyvinylidene fluoride/polymethylmethacrylate fibers grafted with fluoro-contained chemical moieties. The switching of the wettability from hydrophobicity to hydrophilicity was realized by an external electric field. Wei et al. [26] constructed a switchable surface between superhydrophobicity and low adhesion to water or organic liquids through laser texturing technology. These studies achieved the wettability transformation by hydrophilic modification, UV irradiation, pH adjustment, and so on [27,28]. The transformation process always needed a long response time, limiting its practical applications.

We developed the SHI/USOB surface through the liquid-phase deposition (LPD) of SiO_2_ with perfluorooctanoic acid. The surface could achieve the wettability transformation from SHI/USOB properties to SHB/SOI properties through controlling the polarity component on the surface. The coating could be applied for the separation of various immiscible oil–water mixtures and emulsion. The separation performance before and after the wettability transformation has been investigated. The wettability transformation mechanism was also explored. In addition, the long-chain perfluoroalkyl substances could be decomposed by the deposition of the mixture of SiO_2_ and TiO_2_ under UV irradiation, thus reducing the emission of fluorides. The prepared coating provides a feasible method for superwetting surfaces’ fabrication and oil–water separation.

## 2. Materials and Methods

### 2.1. Materials

Ammonium fluotitanate ((NH_4_)_2_TiF_6_), ammonium fluorosilicate ((NH_4_)_2_SiF_6_), boric acid (H_3_BO_3_, ≥99.5%), perfluorooctanoic acid (CF_3_(CF_2_)_6_COOH, PFOA), absolute ethyl alcohol (C_2_H_5_OH), and tetrachloromethane (CCl_4_, 98%) were acquired from Shanghai Macklin Biochemical Co., Ltd. (Shanghai, China). The sponge and soybean oil were purchased from the local market.

### 2.2. Fabrication of Modified Sponge

The sponge (4 × 4 × 2 cm) was rinsed with ethanol and deionized water, and dried in air. Then, 0.1 mol/L of (NH_4_)_2_SiF_6_ and 0.2 mol/L H_3_BO_3_ with equal volumes were stirred. The sponge and 1 g of PFOA were added to the stirred mixture. After stirring at 60 °C for 1 h, the coated sponge was dried in the oven at 40 °C for 2 h. The coated sponge was abbreviated as the SHI/USOB sponge.

During the preparation process, 0.1 mol/L of [(NH_4_)_2_SiF_6_ + (NH_4_)_2_TiF_6_] with the mole ratio of 1:2 was used to prepare the modified sponge based on SiO_2_/TiO_2_ composite nanoparticles, which was abbreviated to SiO_2_/TiO_2_-sponge.

The SHI/USOB sponge was immersed in deionized water, and the wettability of the surface was analyzed every 30 min. After the surface transformed to the SHI/SOI state, the surface was washed with ethanol.

### 2.3. Oil–Water Mixtures Separation

The tetrachloromethane-in-water emulsion was obtained by mixing 2 mL tetrachloromethane, 120 mL water, and 0.32 g Tween-80 under stirring for 3 h. Next, 1 mL water, 114 mL tetrachloromethane, and 0.5 g Span-80 were mixed and stirred for 3 h to prepare a water-in-tetrachloromethane emulsion. The immiscible oil/water mixtures were prepared by coloring the water with Methylene blue and oil with Sudan Ш for observation. The oil flux *F_oil_* was calculated by:(1)$$F_{oil}=V/\left(St\right)$$
where *V* is the oil volume after separation, *S* is the cross-sectional area of the treated sponge to oil/water mixtures, and *t* is the required time for separation. The separation efficiency *η* (%) was calculated by:(2)$$\eta =M_{2}/M_{1}$$
where *M*_1_ and *M*_2_ are oil mass before and after separation, respectively.

### 2.4. Characterization

The micromorphology of the modified sponge was investigated by a Quanta 200 scanning electron microscopy (SEM, FEI, Hillsboro, OR, USA). The chemical components were characterized by a Magna-IR 560 Fourier transform infrared spectroscopy (FTIR, Thermo Nicolet, Madison, WI, USA). The static water contact angles (WCA) were tested five times from different places using a TST-200H optical contact angle meter (Shenzhen testing equipment Co., Ltd., Shenzhen, China).

## 3. Results and Discussion

### 3.1. Characterization of the Modified Surface

During the fabrication process, the mixed (NH_4_)_2_SiF_6_ and H_3_BO_3_ aqueous solution would react [29,30]. The added H_3_BO_3_ reacted with the F^-^ generated from reaction 3 to move the chemical equilibrium reaction 3 to the right, accelerating the exchange reaction of the ligand. Si(OH)_6_^2^^−^ was obtained from the hydrolysis reaction. Finally, SiO_2_ was generated after dehydration, depositing on the sponge surface.
[SiF_6_]^2−^ + n H_2_O ↔ [SiF_6−n_(OH)_n_]^2−^ + n HF(3)
H_3_BO_3_ + 4 HF ↔ BF_4_^-^ + H_3_O^+^ + 2 H_2_O(4)

The SEM images of the original and modified sponge are shown in Figure 1. It was observed that the original sponge showed a flat and smooth surface without any bumps. After modification, SiO_2_ nanoparticles were successfully deposited on the surface of the sponge, constructing a hierarchical structure by the random distribution of particles. The average diameter of SiO_2_ nanoparticles was observed to be 0.4–0.6 µm. The deposited SiO_2_ nanoparticles provided the roughness of the sponge surface.

The XRD analysis of the original and modified sponge is shown in Figure 2. The characteristic peak of SiO_2_ was at 22°, which was observed in the modified sponge. The XRD analysis suggested that the SiO_2_ nanoparticles were successfully loaded on the modified sponge, which played an important role in constituting a hierarchical structure.

The water and oil droplets on the modified sponge are shown in Figure 3. The oil droplets could stand on the SHI/USOB sponge surface under water, while the water droplets wetted on it. Both oil and water droplets wetted the sponge surface when the surface became SHI/SOI. Only water droplets could stand on the SHB/SOI surface. The static CA of the oil droplets under water on the SHI/USOB surface was 153.0°. The static CA of water on SHB/SOI was determined to be 151.8°.

### 3.2. Wettability Transformation Mechanism

The surface morphologies and chemical components of the SHI/USOB and SHB/SOI sponge were analyzed by SEM (Figure 4) and FTIR (Figure 5). As shown in Figure 4, there were still many SiO_2_ nanoparticles attaching to the SHB/SOI surface even after washing with water and ethanol. It could be demonstrated that the LPD method increased the attachment between the sponge and SiO_2_ nanoparticles.

However, the chemical components during the wettability transformation changed a lot. The strong absorption peak between 3000 and 3500 cm^−1^ of the original sponge corresponded to the −NH stretching vibration in −NH_2_. For the SHI/USOB surface, the new absorption peaks at 3200–3000 cm^−1^ were attributed to the stretching vibrations of N-H in NH_4_^+^. The strong absorption peaks at 1402 and 1681 cm^−1^ observed in the SHI/USOB sponge corresponded to the −COO^−^ in PFOA. The stretching vibrations of C-F in PFOA was observed between 1300 and 1100 cm^−1^. Obviously, part of the PFOA connected to SiO_2_ nanoparticles by a dehydration reaction. The other part of the PFOA and the NH_4_^+^ from (NH_4_)_2_SiF_6_ formed the polarity-enhanced fluoride on SiO_2_ nanoparticles. For the SHB/SOI surface, the stretching vibrations of NH_4_^+^ disappeared, which was due to its dissolvable property after submerging in water. The stretching vibrations of C-F in the SHB/SOI sponge were greatly reduced due to the loss of NH_4_^+^. Thus, part of the PFOA was lost after submerging in water and washing with ethanol.

The mechanistic diagram for wettability transformation is presented in Figure 6. The PFOA connected to SiO_2_ nanoparticles was composed of two parts. Part of the PFOA was directly connected. The other part was connected through NH_4_^+^ to form the polarity-enhanced fluoride. The hydrophilic NH_4_^+^ endowed the coating with the function of strong water affinity. Meanwhile, the fluoride groups offered a low surface free energy, resulting in the SHI/USOB surface. The hydrophilic NH_4_^+^ was lost after immersion in water due to its water-soluble property. In addition, the water absorbed on SiO_2_ nanoparticles transformed to −OH groups because of the greater polarity of Si-O bonds. The −OH groups greatly increased the surface energy and caused the modified surface to be SHI/SOI. After washing with ethanol, the −OH groups on the surface were taken away, and part of the PFOA was still connected to SiO_2_ nanoparticles, which made the surface SHB/SOI.

### 3.3. Oil/Water Separation

As shown in Figure 7, the funnel was used as a separation device with the sponge tucked in the upper side. The modified sponge with the ability of wettability transformation could satisfy different types of oil/water mixtures. The SHI/USOB sponge was used to separate the immiscible light oil/water mixture. The separation effect for the soybean oil/water mixture is shown in Figure 7a and Appendix A. The water wetted the sponge, and the soybean oil was blocked on the upper side, which resulted in a complete separation of the soybean oil/water mixture. The separation efficiency and flux of the SHI/USOB sponge for the soybean oil/water mixture were calculated to be 99.4% and 5971 L m^−2^ h^−1^. After the sponge transformed to SHB/SOI, it could be used for heavy oil/water separation. The tetrachloromethane/water mixture was poured into the separation device (Figure 7b and Appendix A). The tetrachloromethane penetrated into the modified sponge, while water was prevented, resulting in a complete separation. The separation efficiency and flux of the SHB/SOI sponge for the tetrachloromethane/water mixture were calculated to be 99.1% and 5347 L m^−2^ h^−1^.

During attempts to separate emulsions, the SHI/USOB sponge could be used for oil-in-water emulsion. The separation performance of the SHI/USOB sponge for tetrachloromethane-in-water emulsion was evaluated. The emulsion was slowly filtered by gravity. The SHI/USOB sponge was wet with the water, falling into the measuring beaker. Finally, the foggy cloudy emulsion became transparent after filtration. As shown in Figure 8a, the oil droplets disappeared completely under an optical microscope. The separation efficiency and flux of the SHI/USOB sponge for tetrachloromethane-in-water emulsion were calculated to be 99.1% and 1517 L m^−2^ h^−1^. The SHB/SOI sponge also had an outstanding performance for separation of water-in-tetrachloromethane emulsion. The water droplets distributed evenly in emulsion were successfully removed after filtration under the optical microscope (Figure 8b). With further evaluation, the separation efficiency and flux of the SHB/SOI sponge for water-in-tetrachloromethane emulsion were 98.5% and 786 L m^−2^ h^−1^. In conclusion, both the SHI/USOB and SHB/SOI sponge exhibited an outstanding separating performance towards various oil/water mixtures, displaying enormous potential in practical usage.

### 3.4. Decomposition of PFOA by Doping of TiO_2_

For fabrication of the SiO_2_/TiO_2_-sponge, the mixed (NH_4_)_2_TiF_6_ and (NH_4_)_2_SiF_6_ were added. The added H_3_BO_3_ reacted with the F^-^ generated from reaction 3 to move the chemical equilibrium reaction 3 and 5 to the right, speeding up the ligand exchange reactions. Ti(OH)_6_^2−^ and Si(OH)_6_^2−^ were obtained from the hydrolysis reaction. Finally, TiO_2_ and SiO_2_ were generated after dehydration, depositing on the sponge surface. The XRD analysis of the SiO_2_/TiO_2_-sponge is shown in Figure 9. The typical diffraction of SiO_2_ was observed at 2θ = 22°. The diffraction peaks at 25.2°, 37.8°, 48.1°, and 53.9° were attributed to the (101), (004), (200), (105) lattice planes of TiO_2_, proving that the synthesized product had a framework of the anatase phase. The XRD analysis suggested that SiO_2_ and TiO_2_ were successfully loaded on the sponge surface.
[TiF_6_]^2−^ + n H_2_O ↔ [TiF_6−n_(OH)_n_]^2-^ + n HF(5)

The wettability of the SiO_2_/TiO_2_-sponge was investigated. As depicted in Figure 10, the water droplets wetted on the SiO_2_/TiO_2_-sponge, while oil droplets could stand on its surface under water with a WCA of 151.1°, which meant that the modified SiO_2_/TiO_2_-sponge was SHI/USOB. Both water and oil droplets could wet the SiO_2_/TiO_2_-sponge after water immersion, illustrating that the SiO_2_/TiO_2_-sponge became SHI/SOI. After ethanol washing, the SiO_2_/TiO_2_-sponge became SHO/SOI with a WCA of 152.6° and oil droplets wetted on it. The SiO_2_/TiO_2_-sponge could also realize the wettability transformation.

The PFOA is widely used for its low surface energy of C-F bond structure. However, the PFOA is difficult to be decomposed for the high electronegativity of F and bond energy of C-F, which will cause environmental pollution and threaten human lives. For the SiO_2_/TiO_2_-sponge, the PFOA absorbed in the form of C_7_F_15_COO^−^ on the modified surface could be oxidized by the holes or ˙OH radicals of the TiO_2_. The SiO_2_/TiO_2_-sponge was placed under the UV light (30 W and 254 nm) with a distance of 10 mm for 3 h. The decomposition process was analyzed by FTIR. As shown in Figure 11, the strong peak between 1300 and 1100 cm^−1^ of the SiO_2_/TiO_2_-sponge corresponded to the C-F stretching vibration, which almost disappeared under UV irradiation. The peaks at 1681 and 1402 cm^−1^ of the SiO_2_/TiO_2_-sponge were attributed to the COO^-^. After UV irradiation, the peak at 1402 cm^−1^ of the SiO_2_/TiO_2_-sponge dropped and the peak at 1681 cm^−1^ disappeared. The FTIR analysis proved the decomposition of PFOA after UV illumination. The sponge after UV illumination could be wetted by water and oil droplets again, meaning that the sponge transformed to the SHI/SOI state again.

## 4. Conclusions

In summary, a new wettability switchable material was fabricated by the liquid-phase deposition method. The wettability of the coating could transform from superhydrophilicity/underwater superoleophobicity (SHI/USOB) to superhydrophobicity/superoleophilicity (SHB/SOI) by controlling the polarity component on the surface. The SHI/USOB surface could be used for the separation of immiscible light oil–water mixtures and oil-in-water emulsions. After the wettability transformation, the SHB/SOI surface could be applied for the separation of immiscible heavy oil–water mixtures and water-in-oil emulsions. The long-chain perfluoroalkyl substances on the surface could be decomposed by mixing SiO_2_ with TiO_2_ nanoparticles under UV irradiation, thus reducing the fluoride emission. The prepared intelligent coating with controlled wettability provides a feasible solution for superwetting material fabrication, which has a bright future in oil–water separation applications.

## Figures and Tables

**Figure 1 nanomaterials-12-03120-f001:**
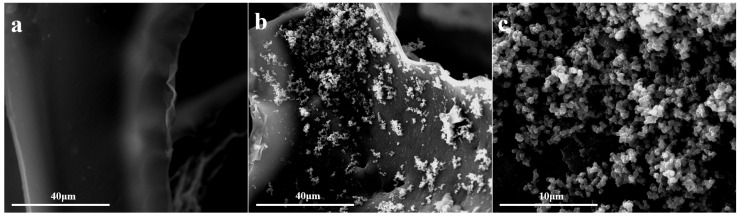
SEM images of sponge before (**a**) and after modification (**b**,**c**).

**Figure 2 nanomaterials-12-03120-f002:**
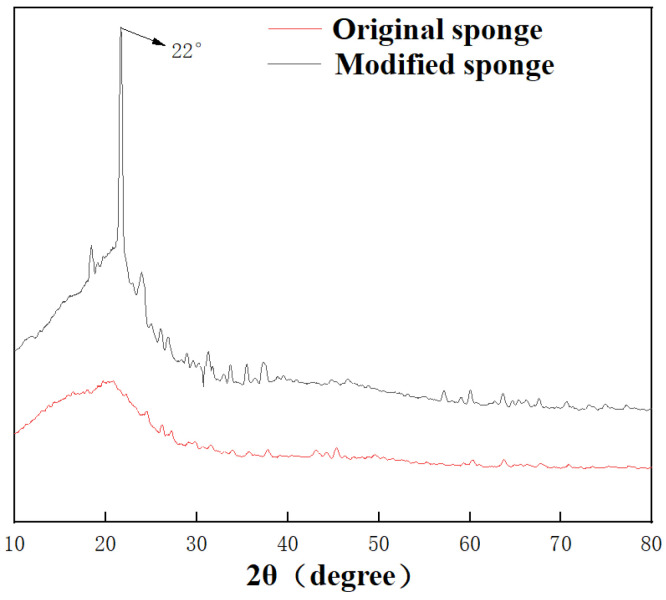
The XRD spectra of original and modified sponge.

**Figure 3 nanomaterials-12-03120-f003:**
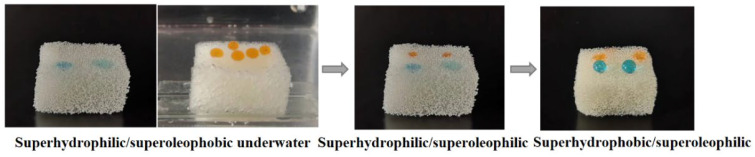
Static wettability of the modified sponge during transformation.

**Figure 4 nanomaterials-12-03120-f004:**
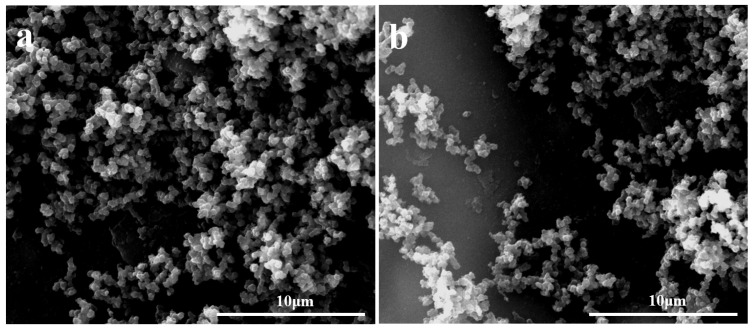
The surface morphologies of the SHI/USOB (**a**) and SHB/SOI (**b**) sponge.

**Figure 5 nanomaterials-12-03120-f005:**
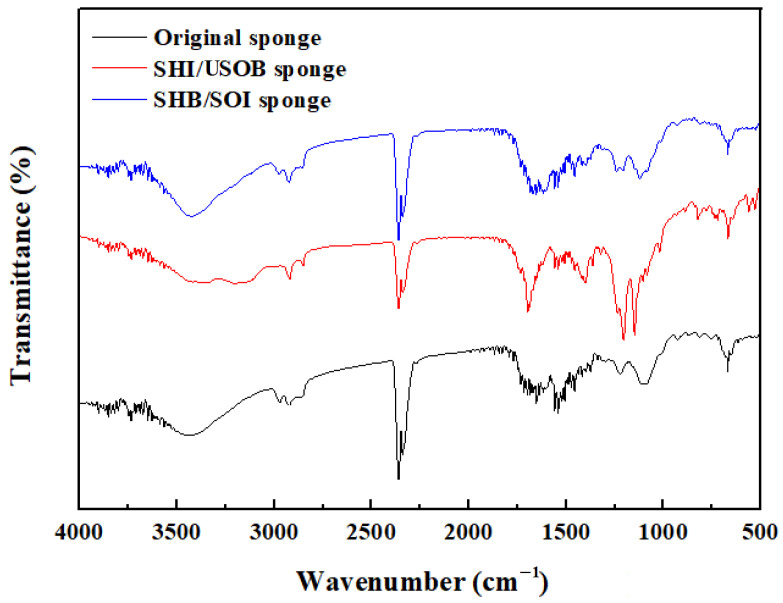
FTIR of the modified sponge before and after wettability transformation.

**Figure 6 nanomaterials-12-03120-f006:**
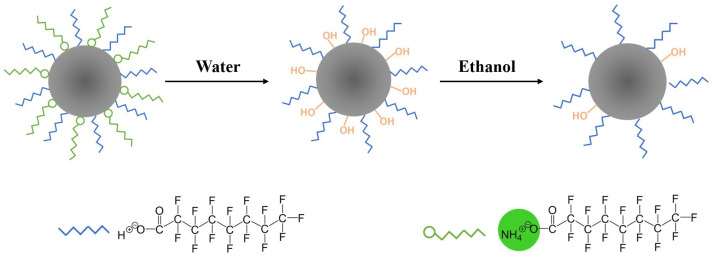
The mechanistic diagram for wettability transformation.

**Figure 7 nanomaterials-12-03120-f007:**
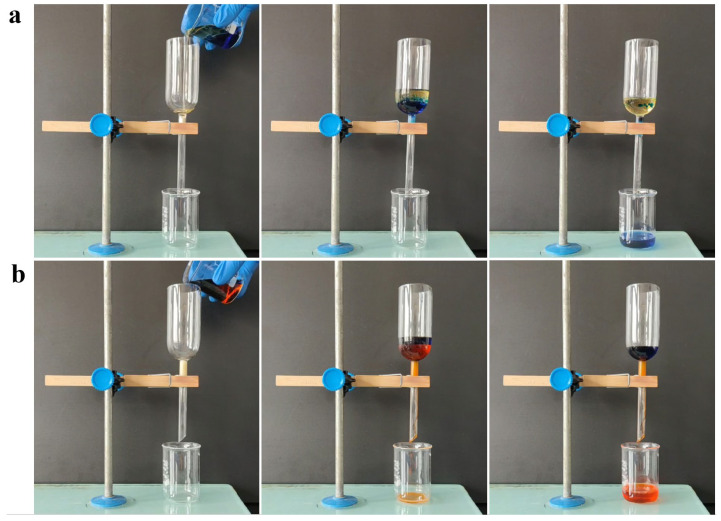
Separation of immiscible soybean oil/water mixture (**a**) and tetrachloromethane/water mixture (**b**).

**Figure 8 nanomaterials-12-03120-f008:**
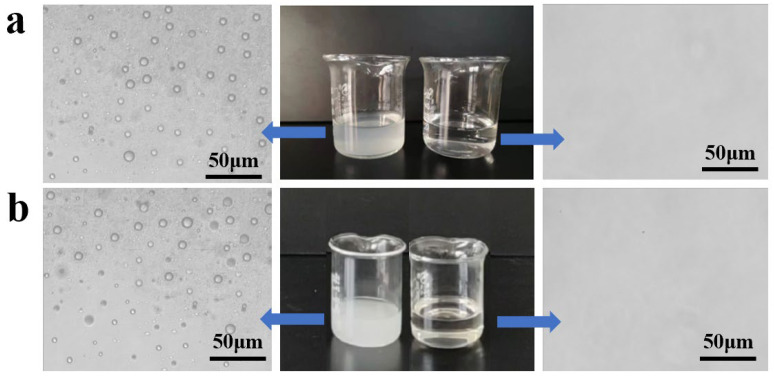
The emulsion under optical microscope before and after filtration. (**a**), Tetrachloromethane-in-water emulsion; (**b**), water-in-tetrachloromethane emulsion.

**Figure 9 nanomaterials-12-03120-f009:**
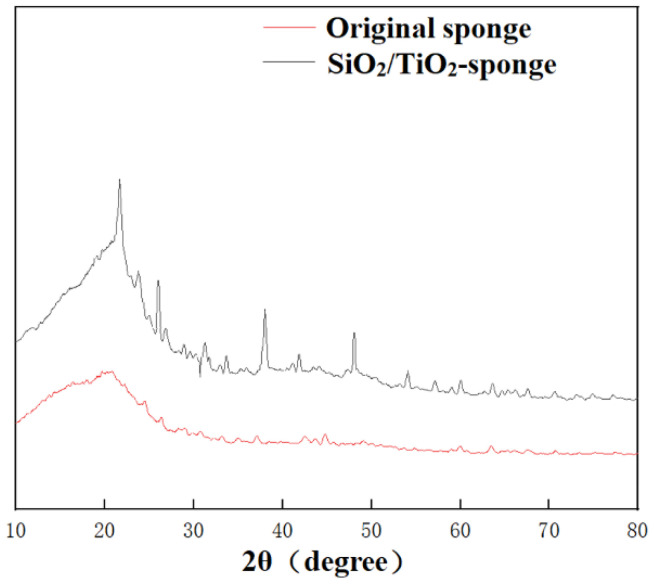
The XRD spectra of original and SiO_2_/TiO_2_-sponge.

**Figure 10 nanomaterials-12-03120-f010:**
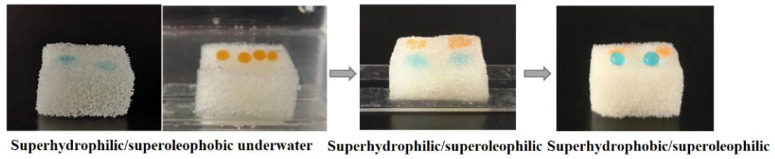
The SiO_2_/TiO_2_-sponge during wettability transformation.

**Figure 11 nanomaterials-12-03120-f011:**
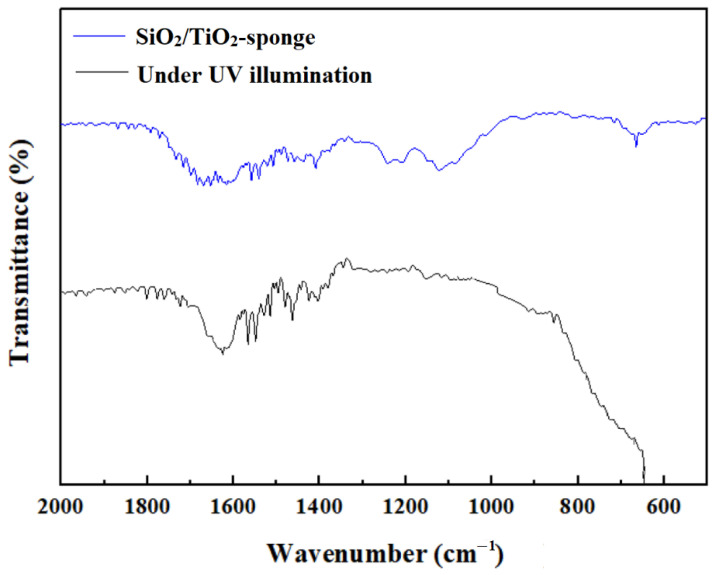
FTIR spectra of SiO_2_/TiO_2_-sponge before and after UV illumination.

## Data Availability

The data presented in this study are available in this article.

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
