# Peer review of "Intelligent Coatings with Controlled Wettability for Oil–Water Separation"

_nanomaterials, 2022, doi:10.3390/nano12183120_

Round 1
Reviewer 1 Report
I recommend this manuscript (nanomaterials-1881471) for publication without any modification.
My opinion is that this manuscript has an interesting subject matter and its compilation is careful.
As far that it is a well written and thought-provoking manuscript with professionally well-supported experiments and results The manuscript is professionally correct.
Author Response
Thank you for your affirmation of our manuscript.
Reviewer 2 Report
The work deals with the actual topic of separation of oil emulsions and purification of aqueous solutions. The authors proposed a very elegant method of switching the hydrophobic/hydrophilic properties of the composite material.
However, the paper requires substantial revision.
1. In the "Introduction" section should be added references to modern actual works on obtaining composites with switchable surface properties.
2. The authors receive in the work nanoparticles based on SiO2/TiO2. In addition to SEM photos, characterization of the obtained particles are not given. The XRD / EDX data for the obtained nanoparticles should be carried out and added.
3. the authors give schemes of chemical processes on page 3. If this is a known optimized method of obtaining, then reference should be made to this method of obtaining. If this method is original, then the analysis data for the processes should be given. How the degree of reaction was determined, how the composition and structure of the products were confirmed, etc. Did boron compounds remain in the final composite?
4. Does the type of substrate/sponge affect the properties of the resulting composite?
5. Characteristic bands in IR spectra indicate a large amount of water. These bands overlap with those for ammonium cations. Did the authors take this fact into account?
After making additions, the article can be submitted to the journal Nanomaterials.
Author Response
Responses to Reviewer 2:
The work deals with the actual topic of separation of oil emulsions and purification of aqueous solutions. The authors proposed a very elegant method of switching the hydrophobic/hydrophilic properties of the composite material.
However, the paper requires substantial revision.
Point 1: In the "Introduction" section should be added references to modern actual works on obtaining composites with switchable surface properties.
Response 1: In the "Introduction" section, more references to modern actual works on obtaining composites with switchable surface properties have been added in line 50-58 in manuscript.
[24] Yang, X.; Jin, H.; Tao, X.; Xu, B.; Lin, S. Photo-switchable smart superhydrophobic surface with controllable superwettability. Polym. Chem. 2021, 12, 5303-5309.
[25] Idriss, H.; Elashnikov, R.; Guselnikova, O.; Postnikov, P.; Kolska, Z.; Lyutakov O.; Švorčík, V. Reversible wettability switching of piezo‑responsive nanostructured polymer fibers by electric field. Chem. Pap. 2020, 75, 191-196.
[26] Wei, D.; Wang, J.; Li, S.; Liu, Y.; Wang, D.; Wang, H. Novel corrosion-resistant behavior and mechanism of a biomimetic surface with switchable wettability on Mg alloy. Chem. Eng. J. 2021, 425, 130450.
Point 2: The authors receive in the work nanoparticles based on SiO2/TiO2. In addition to SEM photos, characterization of the obtained particles are not given. The XRD / EDX data for the obtained nanoparticles should be carried out and added.
Response 2: The XRD analysis of original and modified sponge was added as Figure 2. The characteristic peak of SiO2 is at 22°, which was observed in modified sponge. The XRD analysis suggested that the SiO2 nanoparticles were successfully loaded on modified sponge, which played an important role in constituting a hierarchical structure.
Figure 2. XRD spectra
The XRD analysis of SiO2/TiO2-sponge was shown in Figure 9. The typical diffraction of SiO2 was observed at 2θ= 22°. The diffraction peaks at 25.2°, 37.8°, 48.1° and 53.9° were attributed to the (101), (004), (200), (105) lattice planes of TiO2, proving that the as synthesized product had a framework of anatase phase. The XRD analysis suggested that SiO2 and TiO2 were successfully loaded on sponge surface.
Figure 9. The XRD spectra of original and SiO2/TiO2-sponge.
Point 3: the authors give schemes of chemical processes on page 3. If this is a known optimized method of obtaining, then reference should be made to this method of obtaining. If this method is original, then the analysis data for the processes should be given. How the degree of reaction was determined, how the composition and structure of the products were confirmed, etc. Did boron compounds remain in the final composite?
Response 3: The chemical processes were determined according to the references:
[29] Wang, Y.; Zhang, Y.; Liang, G.; Zhao, X. Fabrication and properties of amorphous silica particles by fluorination of zircon using ammonium bifluoride. J. Fluorine Chem. 2020, 232, 109467.
[30] Dutschke, A.; Diegelmann, C.; Löbmann, P. Preparation of TiO2 thin films on polystyrene by liquid phase deposition. J. Mater. Chem. 2013, 13, 1058-1063.
The references have been added in line 111 in manuscript. The degree of reaction was determined by optimization of the experimental conditions according to the wettability of the modified sponge. The composition and structure of the SiO2 products could be determined by XRD and SEM analysis. The boron compounds were was lost after immersion in water due to its water-soluble property. Thus, the boron compounds could not exist in the final composite.
Point 4: Does the type of substrate/sponge affect the properties of the resulting composite?
Response 4: We used cotton fabrics as the substrate besides sponge during our experiment. The same conclusion could be obtained. We will use biomass as substrates to study the effect of the type of substrate for the properties of the resulting composite in our future research.
Point 5: Characteristic bands in IR spectra indicate a large amount of water. These bands overlap with those for ammonium cations. Did the authors take this fact into account?
Response 5: In IR spectra, the absorption peak of water is between 3600-3000 cm-1, which is very weak and narrow. The stretching vibrations of N-H in NH4+ is between 3200-3000 cm-1, which has two peaks with wide absorption band. Thus, they could be distinguished.

Reviewer 3 Report
Overall, the manuscript reports an interesting story about controlling coating for oil-water separation. It is suggested to be accepted after addressing minor concerns as follows.
1) It seems the SiO2 nanoparticles are not uniformly distributed on the sponge surface, why? Is this design more favorable for the application?
2) Some related works about water treatment are suggested to be considered in the introduction, for example, Mater. Horiz., 2022,9, 1708-1716.
3) Please carefully check the language. Some typos can be found.
Author Response
Responses to Reviewer 3:
Overall, the manuscript reports an interesting story about controlling coating for oil-water separation. It is suggested to be accepted after addressing minor concerns as follows.
Point 1: It seems the SiO2 nanoparticles are not uniformly distributed on the sponge surface, why? Is this design more favorable for the application?
Response 1: The roughness of the surface provided by SiO2 nanoparticles is important for wettability of the modified sponge. Although the SiO2 nanoparticles are not uniformly distributed on the sponge surface, it could still provide a roughness structure on sponge surface. In our future research, we will improve our fabrication method to obtain uniformly distributed SiO2 nanoparticles.
Point 2: Some related works about water treatment are suggested to be considered in the introduction, for example, Mater. Horiz., 2022,9, 1708-1716.
Response 2: The paper about water treatment has been cited in manuscript in line 30 as follows:
[5] Zhang, S.; Xu, X.; Liu, X.; Yang, Q.; Shang, N.; Zhao, X.; Zang, X.; Wang, C.; Wang, Z.; Shapter, J.G.; Yamauchi Y. Heterointerface optimization in a covalent organic framework-on-MXene for high-performance capacitive deionization of oxygenated saline water. Mater. Horiz. 2022, 9, 1708-1716.
Point 3: Please carefully check the language. Some typos can be found.
Response 3: The whole manuscript has been checked to avoid the typos.

Round 2
Reviewer 2 Report
The authors have done a big job for changing the manuscript. The article could be published by Nanomaterials